# Design, Fabrication, and Testing of a Novel 3D 3-Fingered Electrothermal Microgripper with Multiple Degrees of Freedom

**DOI:** 10.3390/mi12040444

**Published:** 2021-04-15

**Authors:** Guoning Si, Liangying Sun, Zhuo Zhang, Xuping Zhang

**Affiliations:** 1School of Medical Instrument and Food Engineering, University of Shanghai for Science and Technology, Shanghai 200093, China; gnsi@usst.edu.cn (G.S.); sly19960721@163.com (L.S.); 2School of Modern Posts, Xi’an University of Posts and Telecommunications, Xi’an 710061, China; zhangzhuozz123@126.com; 3Department of Mechanical and Production Engineering, Aarhus University, 8200 Aarhus, Denmark

**Keywords:** 3D electrothermal actuator, microgripper with multi DOFs, 3D micro-manipulation, three-fingered gripper, zebrafish embryo

## Abstract

This paper presents the design, fabrication, and testing of a novel three-dimensional (3D) three-fingered electrothermal microgripper with multiple degrees of freedom (multi DOFs). Each finger of the microgripper is composed of a V-shaped electrothermal actuator providing one DOF, and a 3D U-shaped electrothermal actuator offering two DOFs in the plane perpendicular to the movement of the V-shaped actuator. As a result, each finger possesses 3D mobilities with three DOFs. Each beam of the actuators is heated externally with the polyimide film. The durability of the polyimide film is tested under different voltages. The static and dynamic properties of the finger are also tested. Experiments show that not only can the microgripper pick and place microobjects, such as micro balls and even highly deformable zebrafish embryos, but can also rotate them in 3D space.

## 1. Introduction

The manipulation of microobjects with high flexibility has long been a hot and seemingly never satisfactory topic for both scientists and engineers [1,2]. Though various micro tools have been developed, these tools, in general, can only perform very simple translational manipulation, i.e., picking up and placing down of microobjects [3,4,5]. Further rotations of microobjects require extensive manual work and skills [6,7,8]. Micropipette [9,10,11], contactless tweezers [12,13], and various microgrippers [14,15,16] proposed in recent decades have been proved to be capable of producing limited 3D mobilities, but not to be good enough to conduct dexterous manipulation of microobject in 3D space. In fact, regardless of the actuation mechanisms, almost all the microgrippers are based on the planar two-jaw design, producing only one DOF [17,18]. The micropipette and contactless tweezers simply provide single DOF and are only suited for manipulating certain objects with limited flexibility. In contrast to the micropipette and contactless tweezers, microgrippers possessing large room for designs have been proven to be promising tools for 3D manipulations. A three-fingered microgripper was proposed, however, each finger only has one DOF [19,20]. A planar microgripper with multi-DOFs has been proposed, but it only brings the flexibility for the planar manipulation of microobjects [21,22]. Over the past decades, the wishes and needs of developing multi-fingered 3D microgripper with multi DOFs have never ended.

In designing microgrippers, choosing a proper actuation mechanism can largely affect the structure and performance of the microgripper. The shape memory alloy (SMA) actuators are usually utilized as actuation of microgrippers with large deformation. However, the SMA actuators have low force and are high cycle time, and are difficult to control because of their thermomechanical nonlinearities [23,24,25]. Electro-static actuators provide aa fast response, but the resulting structure is confined to in-plane design due to fabrication restrictions, and apparently is not suited to 3D microgripper designs [26,27]. In addition, the complicated comb structure limits the room for designing gripping mechanisms with multi DOFs. As a result, nearly all the microgrippers driven by electro-static actuators have two-fingered tweezer-like structures with only one DOF. Electro-magnetic [28,29] and piezo-electric [30,31] actuators often have large size, and hence it is hard to minimize the structure to contain multi actuators for multi DOFs designs. In contrast, electro-thermal actuators produce motion and force using the thermal expansion of beams [32,33,34]. The beams can be with various structures allowing a lot of room for multi-DOF designs and out of plane designs. The V-shaped [35], Z-shaped [36], and U-shaped [37] beams are three most common electro-thermal actuator types among the microgripper designs in literature. However, the V-, Z-, and U-shaped actuators are limited to provide only one DOF and in-plane motion. Thus, the designs of the electro-thermal microgrippers are confined to in-plane design with two gripping jaws, and only have one single DOF.

This work proposes a novel 3D electro-thermal microgripper which can provide multi DOFs with three fingers. Each finger is composed of a novel 3D U-shaped actuator and a V-shaped actuator. The 3D U-shaped actuator and the V-shaped actuator provide two DOFs and one DOF, respectively. The two DOFs provided by the 3D U-shaped actuator are in a plane perpendicular to the motion of the V-shaped actuator, so that each finger will possess 3D motion capability. With three fingers, the 3D microgripper can not only grip microobjects more firmly, but can also rotate them in 3D space. The 3D microgripper is fabricated with 3D printing. The nickel-chromium alloy in the polyimide film is utilized to convert input voltage to thermal energy for heating the beams.

This work is organized as follows. In Section 2, the structure and design of the 3D U-shaped actuator as well as the V-shaped actuator is presented. Section 3 describes the experiment setup and performance testing of the 3D microgripper. Gripping and rotating experiments are demonstrated in Section 4. Finally, summary remarks are given in the conclusion section.

## 2. Structural Design and Fabrication

The schematic structure of the novel 3D electrothermal microgripper is shown in Figure 1a. Each finger of the microgripper consists of a 3D U-shaped electrothermal actuator and a V-shaped electrothermal actuator. The three fingers are identical and configured as depicted in Figure 1b. The structure of the V-shaped actuator integrates the existing designs in the literature, as shown in Figure 1c [38]. The detailed structure of the 3D U-shaped actuators is depicted in Figure 2. Each 3D U-shaped actuator consists of four pairs of beams configured on four sides of the rectangular base. When the polyimide film is used to generate heat and transfer to the pair of beams of the 3D U-shaped actuator, the heated beams on one side cause thermal elongation, thus pushing the unheated beam to bend and produce motion. The deformation of the 3D U-shaped actuator when a pair of slender beams are actuated is shown in Figure 3. The platform of the actuator is fixed, and the end of the actuator is free. It shows that the actuated beams push the unactuated beams to bend. Since there are four pairs of beams configured on the four sides, the tip of the 3D U-shaped actuator can produce a motion in two independent directions, i.e., two DOFs. The advantages of 3D U-shaped actuator is that it has more DOFs with simple operation.

The V-shaped actuator presents a symmetric configuration, where two slanted beams are connected in the shuttle at a certain angle and the inclination of the beams facilitates the in-plane movement. When the polyimide film transfers heat to the beams of the V-shaped actuator, thermal expansion caused by joule heating pushes the shuttle outward, as shown in Figure 3a. The lower base of the 3D U-shaped actuator is fixed to the tip of the V-shaped actuator, and the other end of the 3D U-shaped actuator is connected to the finger extension. The three V-shaped actuators are anchored to a base. The V-shaped actuator has simple structure, and has a large output displacement and output force. Apparently, the two DOFs provided by the 3D U-shaped beams are independent to the motion of the V-shaped actuator. In other words, each finger can produce three DOFs in the 3D space. The 3D microgripper is fabricated using 3D metal printing, as shown in Figure 4a. AlSi10Mg is used as the printing material considering its large thermal conductivity (203 W/(mK)), large thermal expansion coefficient (23 × 10^(−6)/°C), and small Young’s modulus (6 × 10^10 Pa). The dimensions of the 3D microgripper, listed in Table 1, is determined with consideration of fabrication restrictions.

The polyimide electrothermal film is used to provide external heat source to the beams, as shown in Figure 1 and Figure 2. The polyimide film is bonded onto the surface of both the V-shaped beams and the 3D U-shaped beams, as shown in Figure 4a. When a voltage is applied, the polyimide film is heated, and the thermal energy is conducted to the beams causing elongation of the structures. The indirect heating of the beams using the polyimide film allows fabrication of the whole beam structure without insulation problems. The polyimide electrothermal film is in fact a sandwich structure, as shown in Figure 4b. The two outside layers are the polyimide films acting as the insulator, and a nickel-chromium alloy layer is conductive and is used to converts input voltages to thermal energy. The bending stiffness of the polyimide film, the pair of beams of the 3D U-shaped and the beam of V-shaped actuator are 43 N·mm^2^, 5120 N·mm^2^, and 15,000 N·mm^2^ respectively. The bending stiffness of polyimide film is much smaller than that of the pair of beams of the 3D U-shaped and the beam of V-shaped actuator, so the polyimide film can be bend with deformation of the actuator beams and only bring little extra stiffness to the actuator. The resistance of the polyimide film with the size of 6 × 22 mm and 6 × 50 mm is 15.625 Ω and 40 Ω, respectively. It is found that the change of resistance is very weak when applied with a set of voltages [39].

## 3. Experimental Setup and Testing

### 3.1. Experimental Setup

The experimental setup consists of a controllable direct current (DC) power supply, a manual manipulator, the 3D microgripper, and a microscope, as shown in Figure 5a. The 3D microgripper is mounted on a manual manipulator which has three translational motion DOFs. The controllable DC power supply provides input voltages to heat the polyimide electrothermal film. A detailed schematic diagram of the experimental setup is depicted in Figure 5b. The displacement of the microgripper is measured with the microscope. The size of the pixel is calibrated to be 2.792 μm/pixel, and the distance X between the camera and the object is 3 cm, as shown in Figure 6. When measuring the displacement, the object or the microgripper under the microscope should be kept at the distance 3 cm (X) such that each pixel represents 2.792 μm. In the later testing, the template matching is used to measure the tip displacement of the V-shaped and 3D U-shaped actuators, as shown in Figure 7. Figure 7a shows the image of the tip of the microgripper after binarization, and the original image is at the bottom right of Figure 7b.

### 3.2. Durability Testing of the Polyimide Electrothermal Film

Durability of the polyimide electrothermal film is of great importance to the life of the 3D microgripper. In order to test the durability of the electrothermal film, a set of voltages are applied to the polyimide film. First, a voltage is applied to the polyimide film for one minute to heat the polyimide film sufficiently. After cooling down for one minute, another cycle of application of voltage is performed. The sizes of the polyimide films for the V-shaped actuator and the 3D U-shaped actuator are 6 × 22 mm and 6 × 50 mm, respectively. As shown in Figure 8, after 100 cycles of heating and cooling, the output displacements of both the V-shaped actuator and the 3D U-shaped actuator did not decrease too much. The voltages applied to the 3D U-shaped actuator are 15 V, 16 V, and 17 V, and to the V-shaped actuator are 5 V and 6 V, which are exactly below the maximum voltage to avoid burning the polyimide film. Under higher voltages, such as 16 V and 17 V, the displacements of the actuator dropped slightly after many cycles of heating and cooling. For instance, when the voltage is 17 V, after 100 cycles, the displacement decreases from 967.43 μm to 888.37 μm, which decrease by 8.17%. In comparison, under smaller voltages, the output displacements appear to be stable and the actuator can be used for many cycles of operations. For example, when the voltage is 5 V, after 100 cycles, the displacement decreases from 110.84 μm to 108.61 μm, which decrease by 2.01%. The main reason for the decrease of the tip displacement of the 3D microgripper under the voltage of 17 V is due to oxidation or even nearly damaged when being used for many times under high voltages. The high temperature caused by the voltage of 17 V exceeds the melting point of the polyimide film. As a result, the polyimide film is not suited to be used for long-term under high voltages. During the experiment, the voltage is applied to the polyimide film until the displacement of the actuator was stable. Experimental data were collected to calculate the average value as the displacement of the actuator.

The experimental results show that the polyimide film can be used repeatedly for many times without causing obvious decrease in output displacements. This is ideal and satisfactory for micro manipulation applications considering the cheap price (8 yuan) and replaceability.

### 3.3. Static and Dynamic Testing of the 3D U-Shaped Actuator

The static displacement of the 3D U-shaped actuator is tested as shown in Figure 9a. Due to symmetric structure, the output displacements of opposite sides are almost the same. The tip displacements of side 3 and side 4 are larger than the displacements of the side 1 and side 2, which simply is because the flexibilities of the supporting base are different in the two actuating directions. Figure 9a shows the static displacement curve and fitting curve of the four sides of the 3D microgripper. The second order polynomial is well fitted to the measuring data, which is the same as the planar U-shaped actuator case [40]. From the fitting results in Figure 9a, it is found that the tip displacement of the 3D microgripper has a second-order relationship with the voltage. In other words, the output displacement is proportional to square of the input voltage. The experimental results show that the maximum displacements of 3D U-shaped actuator under a voltage of 16 V are 735.13 μm, 723.69 μm, 914.40 μm, and 897.35 μm for the four sides side 1 through 4, respectively. Dynamic responses of the 3D U-shaped actuator are also measured under a voltage of 16 V, as shown in Figure 9b. It shows that the rise time for the 3D U-shaped actuator is approximately 40 s without an overshoot.

### 3.4. Static and Dynamic Testing of the V-Shaped Actuator

The static responses of the V-shaped actuator are tested under different voltages (with increment of 1 V), as shown in Figure 10a. The measured displacement can be fitted using polynomial curve, which agrees to the results in the authors’ previous work [41]. Under a voltage of 6 V, the output displacement of the V-shaped actuator is 178.97 μm. Under a voltage of 6 V, as shown in Figure 10b, it is seen that the output displacement of V-shaped actuator increases rapidly in the beginning 70 s to exceed 120 μm, and slowly approaches to maximum value of 178.97 μm. This shows that the rise time for the V-shaped actuator is approximately 160 s and the displacement reaches stability. The current and the actuator are not very responsive and will be enhanced in closed-loop experiments in the future.

### 3.5. Testing of the U-Shaped Actuator and the V-Shaped Actuator simultaneously

The displacement of the 3D U-shaped actuator and V-shaped actuator of the 3D microgripper driven separately is presented in the above two sub-sections. Now, the displacement of the 3D U-shaped actuator and the V-shaped actuator driven simultaneously is measured. Due to the displacement characteristics of the two opposite sides of the 3D U-shaped actuator are the same, as shown in Section 3.3. Therefore, this section only tests the displacement of two adjacent sides of the 3D U-shaped actuator.

When the voltage of polyimide film on the V-shaped actuator is set to be 6V, we apply different voltages to the polyimide film on the 3D U-shaped actuator and measure the displacement of 3D U-shaped actuator, as shown in Figure 11a. It can be seen from Figure 11a that the displacement of the 3D U-shaped actuator when 3D U-shaped actuator and V-shaped actuator driven simultaneously is almost the same as the result when 3D U-shaped actuator is driven separately. When the voltage of polyimide film on the 3D U-shaped actuator is 16 V, the displacement of the V-shaped actuator is shown in Figure 11b. It can be seen from Figure 11b that the displacement of the V-shaped actuator when the 3D U-shaped actuator and the V-shaped actuator are driven simultaneously is larger than that when the V-shaped actuator is driven separately. The main reason is that the motion principle of the 3D microgripper is thermal expansion, so the 3D U-shaped actuator generates additional displacement along the Z-axis, as shown in Figure 3a. The displacement along the Z-axis when the V-shaped actuator is driven separately, and the 3D U-shaped actuator is driven separately is almost the same as the displacement when the V-shaped actuator and the 3D U-shaped actuator are driven simultaneously.

The experimental results show that the displacement of 3D U-shaped actuator and V-shaped actuator driven at the same time is almost the same, with minor differences, as when driven alone. Though there are mutual thermal effects between the 3D U-shaped actuator and the V-shaped actuator, the effect is small and can be omitted.

## 4. Micro-Manipulation Experiments

With three flexible fingers, the 3D manipulation of microobjects using the fabricated microgripper is demonstrated in this section. The overall flowchart of picking and rotating strategy is depicted in Figure 12. The macro movement of the microgripper is achieved using the manual manipulator. The picking, rotating, and releasing of the microobjects are accomplished with the 3D microgripper. We use micro balls with a diameter of 2 mm and zebrafish embryos with a diameter of around 600~800 μm to illustrate operation of most microobjects. The detailed rotation strategy is illustrated in Figure 13. With the proper collaboration of three fingers, each having three DOFs, the 3D microgripper can rotate a microobject about three independent axes. As shown in Figure 13a, to rotate a microobject about axis Z, the first finger is to move left or right while the other two fingers remain unmoved to hold the object. This is achieved by applying a voltage to the left or right sides of the 3D U-shaped actuator of the first finger. Rotating a microobject about axis X and Y are accomplished by moving one of the fingers in its axial direction. This is achieved by applying a voltage to the V-shaped actuator. At present, the operation experiment of the microgripper on the microobject is open-loop with limited accuracy. Therefore, it is inevitable to cause unintended rotation of the microobject in the operation of the 3D microgripper. For more complicated and flexible rotations, these fingers are required to move collaboratively with precise control. This will be our ongoing research, which is not beyond the focus of this work.

### 4.1. Manipulations of a Micro Ball

In order to clearly observe the rotation of the micro ball during the operation, marking points were made on the surface of the micro ball, and the rotation angle of the micro ball can be calculated according to the movement of the mark points. In addition, there is a white line from the center of the micro ball to the mark point, which is used to indicate the movement state of the micro ball, and it can be seen clearly the movement of the micro ball. As shown in Figure 14, manipulation of the micro ball is divided into several steps as follows. (a) Move the microgripper into field of view; (b) Move the microgripper with the manual manipulator to approach the micro ball; (c) Move the microgripper downward to approach the micro ball; (d) Further move the microgripper to the clamping position; (e) Apply a voltage to the 3D U-shaped actuator and clamp the micro ball; (f) Move the microgripper upwards to show that the micro ball has been firmly picked; (g) Rotate the micro ball about axis Z; (h) Release the micro ball by removing the input voltage. Rotations of the micro ball about other axes are described in Figure 15 and Figure 16, which is achieved by the other fingers of the 3D microgripper. The experimental results show that the micro ball can be clamped and rotated around three axes. In addition, the driving time of the actuator is marked in the figure when the microgripper is operating on the microobject.

### 4.2. Manipulations of a Zebrafish Embryo

In order to clearly observe the rotation of zebrafish embryos during the operation, marking points were made on the surface of the zebrafish embryo, and a white line was drawn along the center of zebrafish embryo to the mark point to describe the movement state of zebrafish embryo. As shown in Figure 17, the manipulation of the zebrafish embryo is divided into several steps as follows: (a) Move the microgripper into field of view; (b) Move the microgripper with the manual manipulator to approach the embryo; (c) Move the microgripper downward to approach the embryo; (d) Further move the microgripper to the clamping position; (e) Apply a voltage to the 3D U-shaped actuator and clamp the embryo; (f) Move the microgripper upwards to show that the embryo has been firmly picked; (g) Rotate the embryo about axis Z; (h) Release the embryo by removing the input voltage. Rotations of the embryo about other axes are described in Figure 18 and Figure 19.

The rotation angle of the microobject caused by the 3D microgripper is calculated according to the position change of the marked points on the microobject, as shown in Figure 20. The arc length of the microobject is calculated by *L* = *X*_2_ − *X*_1_, and then the rotation angle *N* of the microobject is calculated by $N=\frac{180L}{\pi R}$. The actual rotation angle is shown in Table 2. The rotation angle of the microobject can be further improved by choosing the appropriate tip material to enhance the friction or precisely control the interaction between the tip and the microobject.

## 5. Conclusions

In this work, we have designed and fabricated a novel 3D electrothermal microgripper with three fingers. Each finger is composed of a V-shaped actuator and a 3D U-shaped actuator, which provides one and two DOFs in 3D space respectively. The 3D microgripper is fabricated using 3D metal printing and the polyimide film is utilized as the external heat source. Experimental testing has been performed to demonstrate satisfactory durability of the microgripper. In addition, static and dynamic performances have also been tested. With designed operation methods, both the micro ball with a diameter of 2 mm and the zebrafish embryo with a diameter of around 600~800 microns can be handled freely. Furthermore, with the collaboration of three fingers, the microgripper can rotate the microobjects in 3D space. Based on the 3D microgripper, our ongoing work focuses on the controlling, force sensing, and 3D micro visions.

## Figures and Tables

**Figure 1 micromachines-12-00444-f001:**
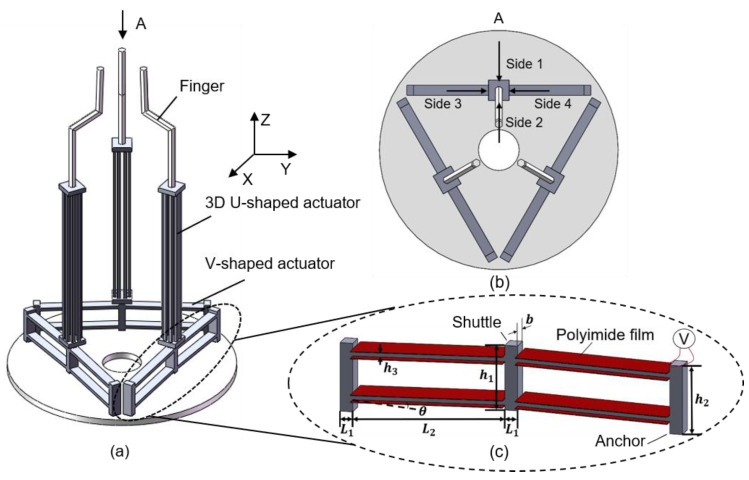
The structure of the 3D electrothermal microgripper. (**a**) Structure of the 3D electrothermal microgripper; (**b**) Top view of the 3D microgripper; (**c**) The V-shaped electrothermal actuator.

**Figure 2 micromachines-12-00444-f002:**
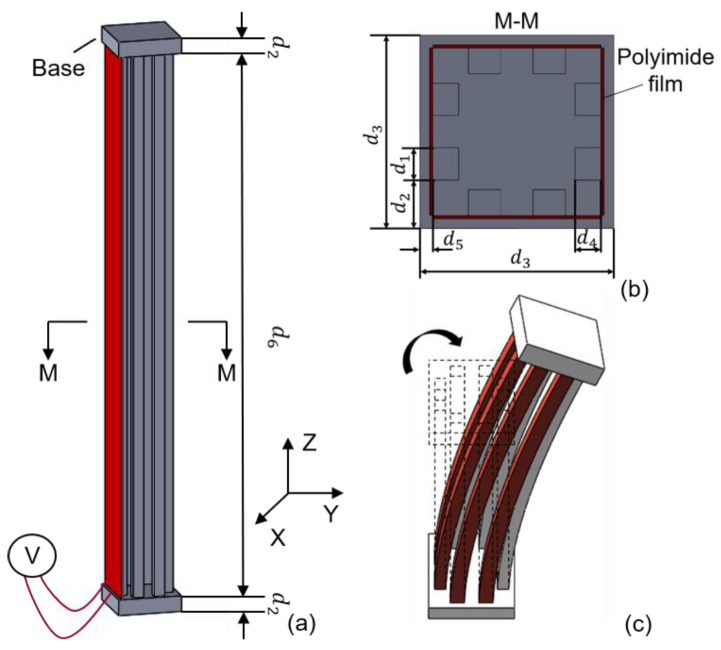
The structure of the 3D U-shaped electrothermal actuator. (**a**) Structure of the 3D U-shaped electrothermal actuator; (**b**) Sectional view of 3D U-shaped electrothermal actuator; (**c**) The 3D U-shaped actuator in bending.

**Figure 3 micromachines-12-00444-f003:**
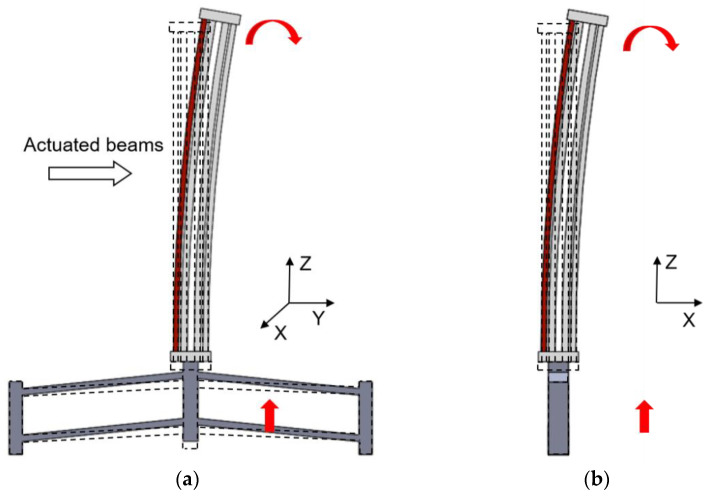
Deformation of the 3D U-shaped and V-shaped actuator. (**a**) The 3D U-shaped actuator bends to axis-Y and V-shaped actuator bends to axis-Z; (**b**) The 3D U-shaped actuator bends to axis-X and V-shaped actuator bends to axis-Z.

**Figure 4 micromachines-12-00444-f004:**
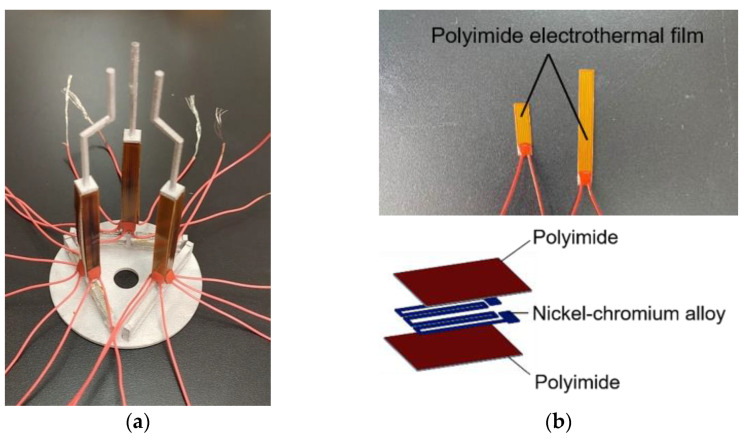
The fabricated 3D microgripper with the polyimide film. (**a**) The fabricated 3D microgripper; (**b**) The polyimide electrothermal film.

**Figure 5 micromachines-12-00444-f005:**
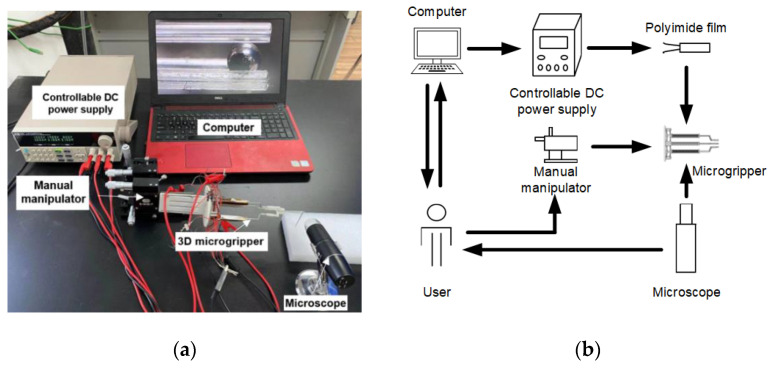
Experimental setup. (**a**) Experimental setup; (**b**) Schematic diagram of the experimental setup.

**Figure 6 micromachines-12-00444-f006:**
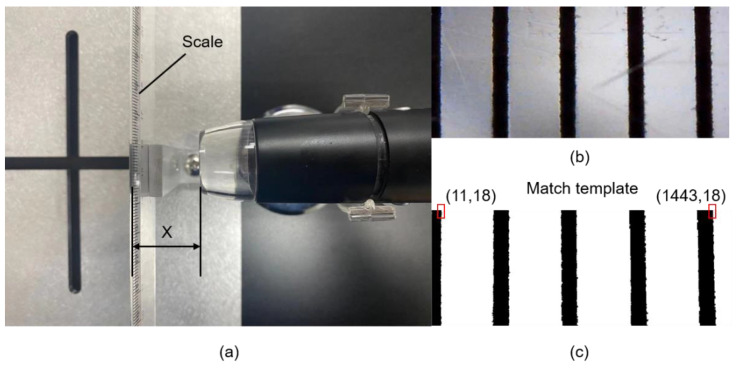
Calibration the actual pixel size. (**a**) The equipment and location for calibration; (**b**) Color image of the scale in the microscope; (**c**) The binarization image of the scale.

**Figure 7 micromachines-12-00444-f007:**
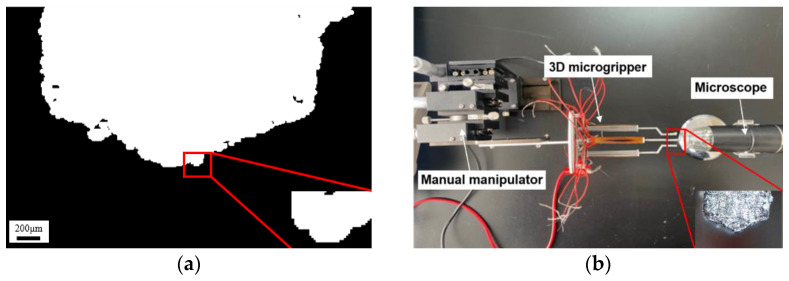
Template matching of the microgripper tip. (**a**) The matching template; (**b**) displacement measuring.

**Figure 8 micromachines-12-00444-f008:**
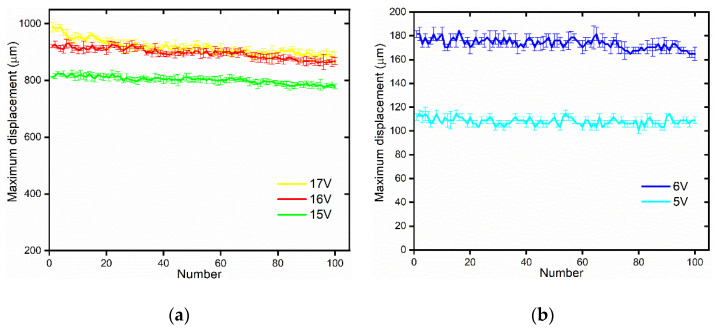
Durability testing of the polyimide electrothermal film. (**a**) Polyimide film with size of 6 mm × 50 mm; (**b**) Polyimide film with size of 6 mm × 22 mm.

**Figure 9 micromachines-12-00444-f009:**
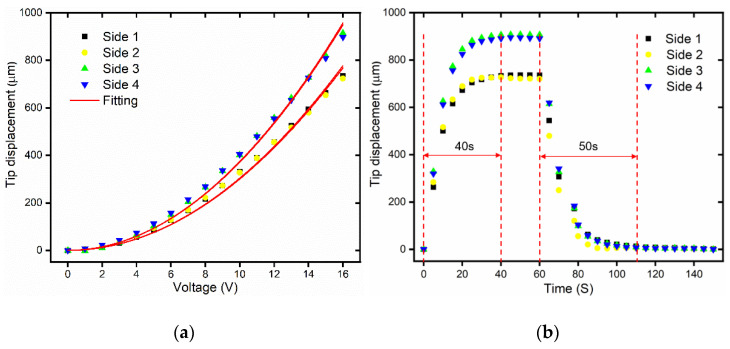
The static and dynamic testing of the 3D U-shaped actuator. (**a**) The static displacements vs. voltages; (**b**) The dynamic displacement under a voltage of 16 V.

**Figure 10 micromachines-12-00444-f010:**
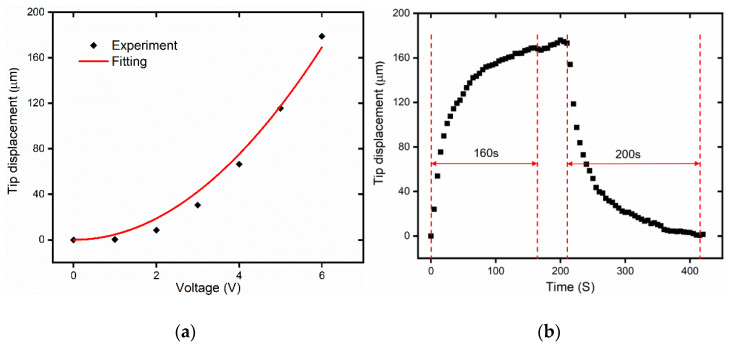
The static and dynamic testing of the microgripper V-shaped actuator. (**a**) The static displacements vs. voltages; (**b**) The dynamic response of the V-shaped actuator under a voltage of 6 V.

**Figure 11 micromachines-12-00444-f011:**
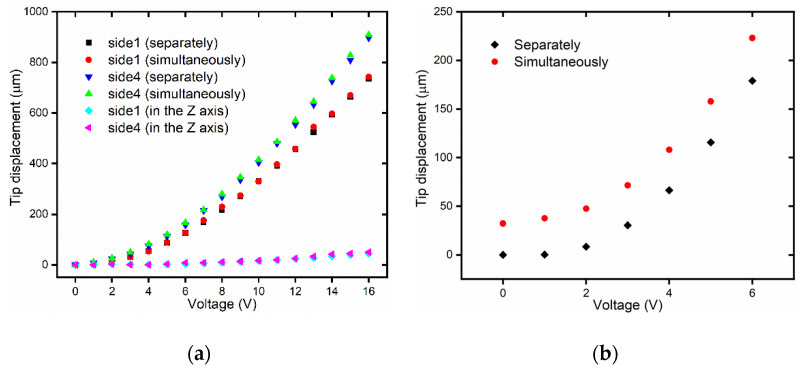
The displacement when V-shaped actuator and 3D U-shaped actuator move simultaneously and separately; (**a**) The displacement of the 3D U-shaped actuator; (**b**) The displacement of the V-shaped actuator.

**Figure 12 micromachines-12-00444-f012:**
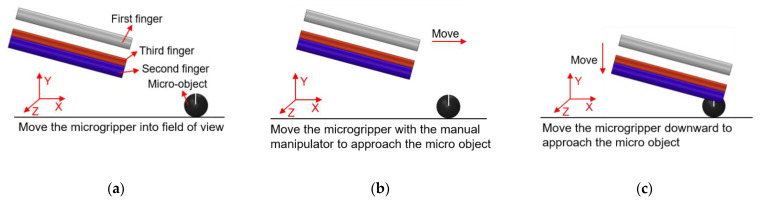
The schematic diagram of micro-objects manipulations.

**Figure 13 micromachines-12-00444-f013:**
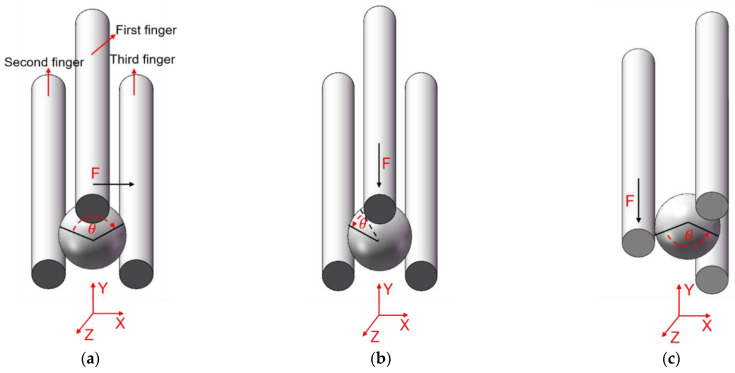
Rotations of micro an object with the microgripper: (**a**) Rotation about axis Z, (**b**) Rotation about axis X, and (**c**) Rotation about axis Y.

**Figure 14 micromachines-12-00444-f014:**
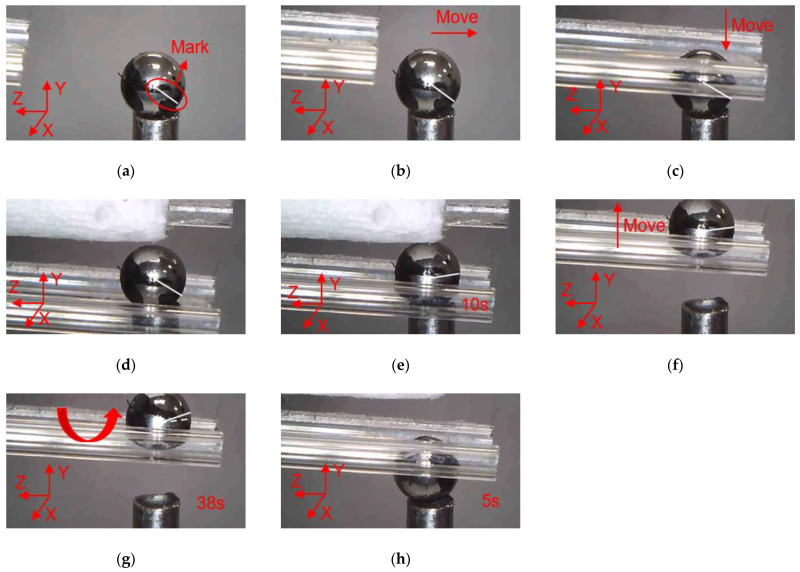
Manipulation of a micro ball: (**a**) Initial state; (**b**) Microgripper is moved to approach the micro ball; (**c**) The microgripper is moved downward to approach the micro ball; (**d**) The microgripper is reached to the clamping position; (**e**) The micro ball is clamped; (**f**) The micro ball is move upward; (**g**) The micro ball is rotated about axis Z; (**h**) The micro ball is released.

**Figure 15 micromachines-12-00444-f015:**
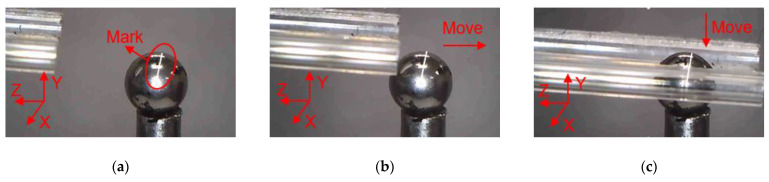
Manipulation of a micro ball: (**a**) Initial state; (**b**) Microgripper is moved to approach the micro ball; (**c**) The microgripper is moved downward to approach the micro ball; (**d**) The microgripper is reached to the clamping position; (**e**) The micro ball is clamped; (**f**) The micro ball is move upward; (**g**) The micro ball is rotated about axis X; (**h**) The micro ball is released.

**Figure 16 micromachines-12-00444-f016:**
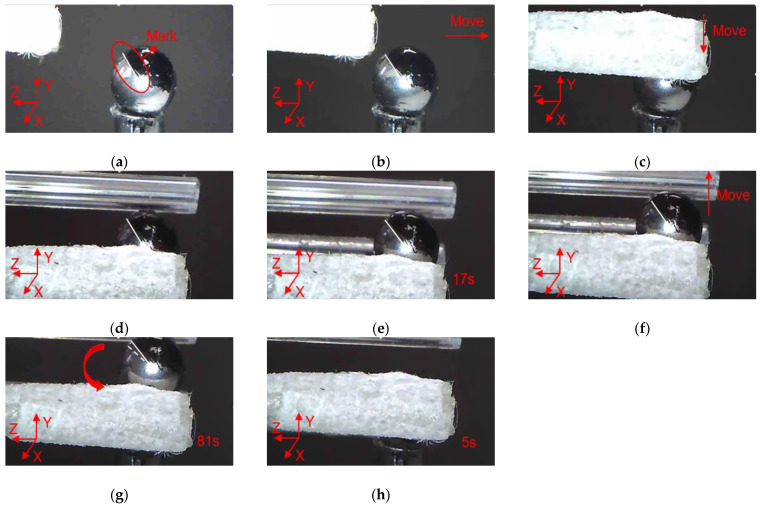
Manipulation of a micro ball: (**a**) Initial state; (**b**) Microgripper is moved to approach the micro ball; (**c**) The microgripper is moved downward to approach the micro ball; (**d**) The microgripper is reached to the clamping position; (**e**) The micro ball is clamped; (**f**) The micro ball is move upward; (**g**) The micro ball is rotated about axis Y; (**h**) The micro ball is released.

**Figure 17 micromachines-12-00444-f017:**
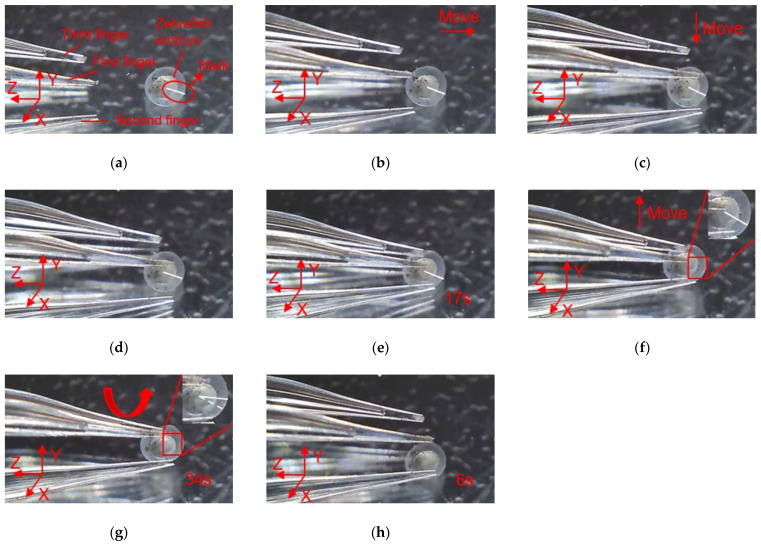
Manipulation of a embryo: (**a**) Initial state; (**b**) Microgripper is moved to approach the embryo; (**c**) The microgripper is moved downward to approach the embryo; (**d**) The microgripper is reached to the clamping position; (**e**) The embryo is clamped; (**f**) The embryo is move upward; (**g**) The embryo is rotated about axis Z; (**h**) The embryo is released.

**Figure 18 micromachines-12-00444-f018:**
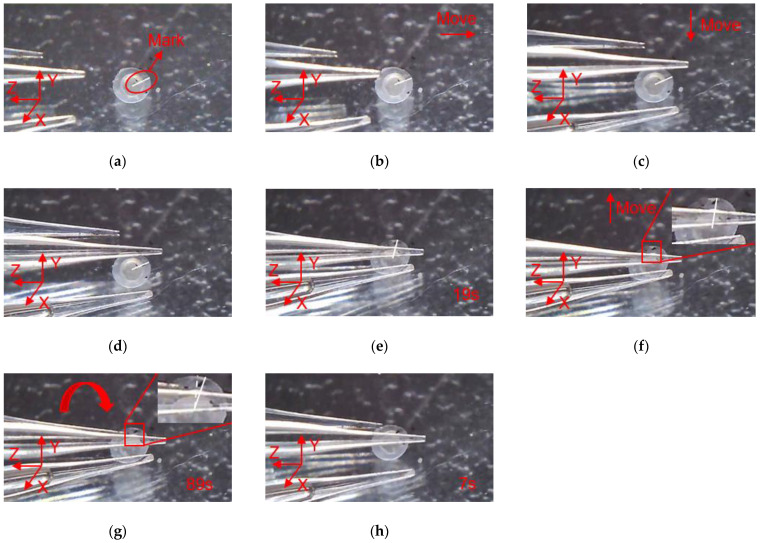
Manipulation of a embryo: (**a**) Initial state; (**b**) Microgripper is moved to approach the embryo; (**c**) The microgripper is moved downward to approach the embryo; (**d**) The microgripper is reached to the clamping position; (**e**) The embryo is clamped; (**f**) The embryo is move upward; (**g**) The embryo is rotated about axis X; (**h**) The embryo is released.

**Figure 19 micromachines-12-00444-f019:**
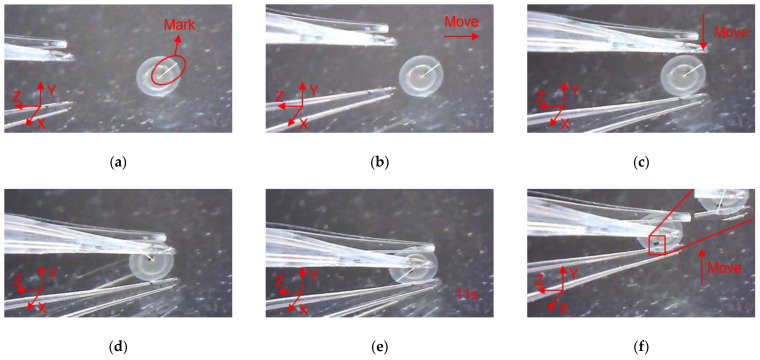
Manipulation of a embryo: (**a**) Initial state; (**b**) Microgripper is moved to approach the embryo; (**c**) The microgripper is moved downward to approach the embryo; (**d**) The microgripper is reached to the clamping position; (**e**) The embryo is clamped; (**f**) The embryo is move upward; (**g**) The embryo is rotated about axis Y; (**h**) The embryo is released.

**Figure 20 micromachines-12-00444-f020:**
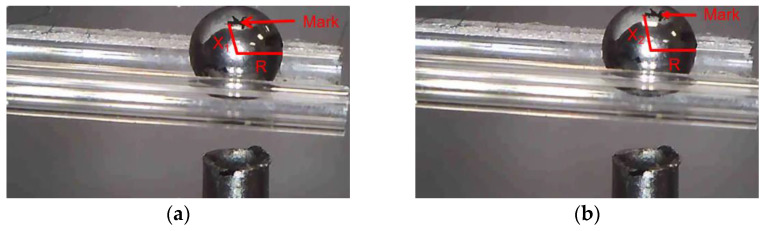
Before and after the rotation of the micro ball. (**a**) Before the micro ball rotates. (**b**) After the micro ball rotates.

**Table 1 micromachines-12-00444-t001:** Dimensions of the 3D Microgripper (Unit of Length: mm).

Symbol	Actuator	Definition	Value
d_1_	3D U-shape actuator	Width of beam	1
d_2_		The distance between beam and base boundary	1.5
d_3_		Length of base	6
d_4_		Thickness of beam	0.8
d_5_		The distance between beam and base boundary	0.4
d_6_		Length of beam	52
b	V-shaped actuator	Thickness of beam and shuttle	3
h_1_		Length of shuttle	10.26
h_2_		Length of anchor	10.86
h_3_		Thickness of beam	1
L_1_		Width of shuttle and anchor	2
L_2_		Span of the beam	24
θ		Inclination of beam	1.89°

**Table 2 micromachines-12-00444-t002:** Rotation angle of the micro-object in experiment.

	Rotation about Axis X	Rotation about Axis Y	Rotation about Axis Z
The angle of rotation of the ball	13.35°	16.21°	15.60°
The angle of rotation of the zebrafish embryo	21.61°	20.46°	23.70°

## Data Availability

The data presented in this study are available inside the main text.

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
