# Peer review of "Design, Fabrication, and Testing of a Novel 3D 3-Fingered Electrothermal Microgripper with Multiple Degrees of Freedom"

_micromachines, 2021, doi:10.3390/mi12040444_

Round 1

Reviewer 1 Report

The authors wrote "The polyimide film is utilized to convert input voltage to thermal energy thus heating the beams.", however it is incorrect. The electrical energy is converted to thermal energy by Ni-Cr alloy.

The authors describe them as U-shaped and V-shaped, but it is difficult to tell from Figures 1 and 2 how U and V-shaped actuators are. The reviewer suggests to explain details and merits of U and V-shaped actuators for the clearer understanding of readers.

Why did the authors use Ni-Cr as a heat source? The flexibility of this part greatly affects the performance of the actuator.

SMA is also widely used as a similar micro actuator, but what are its advantages over SMA? Please compare with previous literatures.

The reviewer could not figure out what the picture in Figure 7(a) represented. What is the white part of the image that has been binarized? Please include the original image.

From the results in Fig. 8, it appears that the actuation is not very stable. Since it is a very fine actuation, i. the effect of wind (such as from air conditioner) is eliminated, and ii. the resolution of the measured image should be confirmed to be sufficient.

The reviewer asks to write in the text the physical meaning of the polynomial fitting in Figure 9 may be written in the reference [35].

The authors describe only the voltage in the paper, but I think it is necessary to describe the resistance or current value because it is impossible to imagine the temperature without knowing the power.

From the durability test, it appears that the drive is not very stable, but does it move in exactly the same direction symmetrically as the authors insist? In the reviewer's experience, it may move asymmetrically due to hysteresis.

Are there any mutual heat effects when nearby actuators are activated?

The authors got the 3D movement from the 2D photo, but will not that cause an error for the actual 3D displacement?

The frictional asymmetry of the finger is used to rotate it to X, Y, and Z, respectively, but it is not possible to control it to X, Y, and Z at the same time, so we consider it a weak degree of freedom.

Unintended rotation can be seen at the timing shown in Figure 14(e). Isn't it possible that the degree of freedom is reduced because the operations are not independent?

One of the drawbacks of thermal actuators is that they are not very responsive. Is it possible to write down the time for each demonstration photo so that we can understand the responsiveness?

Reviewer 2 Report

An excellent paper on a timely subject.  Please review the spellings/grammar and structure of the paper prior to submitting the final paper.

Reviewer 3 Report

The authors present a microgripper system based on a 3D printed metal scaffold that is actuated by a number of individually controlled polyimide film components. These locally heat the structure and induce deformations that result in controlled deformation at the microgripper tips. Uniquely, this structure incorporates multi-modal actuation, with different actuator types resulting in multiple degrees of freedom for each of three actuator tips. This permits the rotational motion of samples, including biological ones, along x,y and z axes despite fixing the base in a given position. I am supportive of this manuscript for publication, though I recommend that the authors address my comments to improve the clarity of their manuscript.

  • Figure 7 should have scale bars

  • Figure 8 vertical axis should read (µm) rather than /µm, since the latter could give the impression that the units here are in 1/µm (inverse microns).

  • Figure 8 caption should specify that this is the maximum displacement

  • Regarding any long-term drift, is there any change in the settling, 0V unactuated position? Is this potentially the cause of the drift in the tip displacement in the 17V case in Figure 8?

  • Line 176; is a second order polynomial the best physics-based function to fit to this data? That is, this may produce a decent R^2 fit, but is there literature that suggests a polynomial relationship between actuator displacement and applied voltage? Or would this better be described physically by a power law fit?

  • Is there a reason a different plotting format is used between figure 10a and figure 11?

  • This manuscript would benefit significantly from SI videos showing the actuation of each of these fingers and the manipulation of objects.

  • Various English errors, e.g.:
    • Line 45, should be resulting rather than resulted
    • Line 158, should be dropped rather than drop
    • Check the rest of the text to make sure that the language is as intended.

Round 2

Reviewer 1 Report

>>We have made the modifications on Figure 2 and Figure 3 to better illustrate the working principle of both the 3D U-shaped actuator and the V-shaped actuator.

The reviewer would like to know the design and the working principle of each U-shaped and V-shaped actuator. It is possible that the names U-shaped and V-shaped are not common within the field of actuators, therefore it is better to explain them.

>> it can bend with deformation of the actuator beams without bringing extra stiffness to the actuator.

“Without bringing” may be an exaggeration. It would be good to mention the bending stiffness of the material.

>> The shape memory alloy (SMA) actuators are usually utilized as actuation of microgrippers with large deformation, however the output displacements are discontinuous which obviously is not suited to the development of microgripper.

What do you mean by "discontinuous"? Also, the reviewer thinks "obviously" is an overstatement. The reviewer thinks the need for feedback control is the same for both thermal actuators and SMA.

>> This time we have collected a set of data of output displacement and averaged them to represent the steady output displacement, and the updated Figure 8 is as follows. It is seen that the curve is more stable.

If the authors took multiple averages, the authors would need to describe the data analysis in the text. The authors may want to split the graph and add the error bars. Also, the displacement is not sufficiently visible to see if it is being driven, which makes the reviewer wonder if the results are appropriate for a durability test.

>> It is found that the resistance does not change when applied with a set of voltages.

The reviewer would think that heating a conductor would increase its resistance, but why did not the resistance change? Do you use any specific materials?

>> The rise time of the electrothermal actuators is a little bit indeed large and this can be further improved via feedback controls, which is one of our ongoing research focuses.

Feedback control can increase the accuracy, but the rise time can be shortened by simply increasing the input energy. The reviewer suggested inserting the drive time into the photo, is it difficult to add?

Round 3

Reviewer 1 Report

It would be better to explain the number of seconds written in the picture in the caption of the figures.